# Weigh-in-Motion Site for Type Approval of Vehicle Mass Enforcement Systems in Poland

**DOI:** 10.3390/s23229290

**Published:** 2023-11-20

**Authors:** Janusz Gajda, Ryszard Sroka, Piotr Burnos, Mateusz Daniol

**Affiliations:** Department of Measurement and Electronics, AGH University of Krakow, Al. A. Mickiewicza 30, 30-059 Krakow, Poland; jgajda@agh.edu.pl (J.G.); burnos@agh.edu.pl (P.B.); daniol@agh.edu.pl (M.D.)

**Keywords:** direct mass enforcement, WIM systems, type approval, metrological examination, accuracy of WIM systems

## Abstract

The need to protect road infrastructure makes it necessary to direct the mass enforcement control of motor vehicles. Such control, in order to fulfil its role, must be continuous and universal. The only tool currently known to achieve these goals are weigh-in-motion (WIM) systems. The implementation of mass enforcement WIM systems is possible only if the requirements for their metrological properties are formulated, followed by the implementation of administrative procedures for the type approval of WIM systems, rules for their metrological examination, and administrative regulations for their practical use. The AGH University of Krakow, in cooperation with the Central Office of Measures (Polish National Metrological Institute), has been conducting research in this direction for many years, and, now, as part of a research project financed by the Ministry of Education and Science. In this paper, we describe a unique WIM system located in the south of Poland and the results of over two years of our research. These studies are intended to lead to the formulation of requirements for metrological legalisation procedures for this type of system. Our efforts are focused on implementing WIM systems in Poland for direct mass enforcement. The tests carried out confirmed that the constructed system is fully functional. Its equipment with quartz and bending plate load sensors allows for the comparison of both technologies and the measurement of many parameters of the weighed vehicle and environmental parameters affecting weighing accuracy. The tests confirmed the stability of its metrological parameters. The GVW maximal measurement error does not exceed 5%, and the single axle load maximal measurement error does not exceed 12%. The sensors of the environmental parameters allow for the search for correlations between weighing accuracy and the intensity of these parameters.

## 1. Introduction

The functioning and development of modern societies is linked to their ability to move around quickly and safely, as well as their ability to transport goods by land. The expansion and maintenance of the road network is undoubtedly a major challenge in an era of intensive automotive development. The road network in any country is one of its largest and most expensive investments. Equally costly are the consequences of not ensuring road safety. The need to protect road infrastructure from destruction and accelerated wear and to ensure traffic safety is therefore obvious. The need to protect the conditions of fair competition in transport is also clear in this context. Therefore, there arises a need for the effective and automatic detection of overloaded vehicles, which are one of the main causes of the adverse phenomena mentioned above, and their elimination from road traffic.

One of the tools for measuring the weight and axle load of individual vehicles involved in road traffic and detecting overloaded vehicles are weighing systems for vehicles in motion (WIM). These systems are sometimes referred to as high-speed weigh-in-motion (HS-WIM) systems to distinguish them from low-speed systems (LS-WIM), in which the permissible speed of the weighed vehicle is of the order of 5 km/h. LS-WIM systems are currently used as scales for administrative purposes.

The accuracy of the WIM system is influenced by several factors such as the speed of vehicles passing through the system, the weighing system configuration (e.g., number of sensors), and the required quality of the pavement in the station area. Environmental factors also have a significant impact on the accuracy of weighing. In addition, the influence of these factors depends on the technology used in the construction of the pavement and the type of load sensors used [1,2].

WIM systems aim to measure the dynamic load that the wheels of a moving vehicle exert on the pavement. Based on this measurement, the systems then estimate the sought values of the static load of the individual axles and the gross weight of the vehicle [3].

Each WIM system consists of load sensors mounted in the pavement, placed in one, two or even several lines (multi-sensor WIM systems), perpendicular to the direction of traffic flow [4,5]. The load sensors cooperate with electronic conditioning systems, a system for acquiring measurement data, and a computer system that executes an algorithm to estimate the vehicle’s axle static load and gross weight.

The conditions under which vehicles are weighed in WIM systems determine the accuracy of the results obtained [6]. The dominant role is played by the sensors, which respond to dynamic load due to the movement of the vehicle. Thus, the speed of the vehicle, the condition and mechanical parameters of its suspension, the quality of the pavement, the driving style, atmospheric factors, and among these, most significantly, the temperature and the direction and strength of the wind have an impact [7,8,9,10,11,12,13].

These factors contribute to the measurement uncertainty of WIM systems, particularly those with only two load sensor lines. Consequently, currently, WIMs only function as preselection systems that assist administrative static weighing procedures.

At the same time, the problem of the control of the weighing of vehicles in an effective way requires an urgent solution in Poland. This necessity stems, inter alia, from the conclusions of a report drawn up by the Supreme Audit Office (NIK) [14] and from data published by the Road and Bridge Research Institute [15]. The basic conclusion from the published data is that every third truck moving on the roads in Poland is overloaded. This is confirmed by the weighing results of five-axle vehicles (two-axle tractor + three-axle trailer) collected by the authors (Figure 1), and it is clear that vehicles exceeding the permissible axle load and the permissible gross weight are considered to be the main factor affecting the deterioration of roads and bridges.

Normative documents on the automatic weighing of motor vehicles are being developed by international metrological organisations [16,17], as well as in individual countries, taking into account the applicable legal regulations [18,19,20]. These actions show that the problem of effectively eliminating overloaded vehicles from traffic is recognised in many countries.

The solution to eliminating most of the shortcomings of the current vehicle weight control system mentioned in the cited NIK report is the introduction of WIM systems operating in the direct mass enforcement mode. The Ministry of Economic Development and Technology (Poland) has obliged the Central Office of Measures to develop the requirements that the enforcement WIM systems should meet and to determine the scope of the tests carried out during legal metrological control of such systems. Approval for the administrative use of the enforcement WIM systems requires the approval of the type of systems offered on the market, and their further use requires periodic metrological legalisation.

In this paper, the structure and parameters of the investigated WIM system are presented. The purpose of its construction was to provide conditions for long-term research on the influence of various factors interfering with weighing, such as meteorological factors, vehicle speed, load sensor technology, and manoeuvres performed by the vehicle during the passage through the WIM station. The results of these studies will be used in the formulation of requirements for administrative WIM systems and in the design of metrological testing procedures for such systems used during type approval, initial legalisation, and re-legalisation.

There are known publications describing the various configurations and equipment of WIM systems. The goals of their construction and research are also different. In Section 2, the review of the literature from this point of view is presented.

The work is organised as follows: In Section 2, the literature review is presented, Section 3 describes the construction and equipment of the WIM site, Section 4 presents the results of the tests conducted to assess the quality of the pavement. Section 5 contains the sample results of the measurements carried out at the described WIM site. In Section 6, the results of the tests aimed at determining the accuracy of the results of weighing vehicles at the described WIM site are discussed. The accuracy was assessed separately for static axle load and gross vehicle weight (GVW) measurements. Section 7 is a summary of the article.

## 2. Literature Review

WIM systems are built using many different sensors and their mutual configuration. This depends on the purpose of the system being built, its expected accuracy, and takes into account the economic aspect. The WIM system consists of four lines of quartz load sensors and four piezoelectric sensors, installed obliquely to the direction of vehicle movement, as is presented in [21]. The load sensors were divided into two grids A and B, each containing two lines of load sensors. Grid A is used to estimate the weighing result, and grid B is used to verify its correctness. The system is equipped with only two pavement temperature sensors. An interesting aspect of the WIM system is that it is equipped with 56 strain gauge sensors [22]. The sensors were installed in four lines (perpendicular to the direction of vehicle movement), with 14 sensors in each line. The load sensors operate in a half-bridge system with temperature compensation. Additionally, the system is equipped with two induction loops at both ends of the sensor system. Data acquisition is triggered by a signal from the first loop and is ended with a signal from the second loop. The system is not equipped with sensors enabling the measurement of environmental parameters. The authors of [23] tried to find an answer to the question of how to improve the accuracy of multi-sensor WIM systems while maintaining costs at an acceptable level and presents an analysis of the various configurations of load sensors in the WIM system. A WIM system with a configuration of load and additional sensors that enables the detection of incorrect passages through the WIM station, i.e., passages with a change in the vehicle’s trajectory or speed, was analysed in [24]. The system is equipped with strain gauge load sensors, induction loops, and piezoelectric sensors for, among other functions, the detection of twin wheels and the determination of where the vehicle passes through the station. This makes it possible to take into account the variable sensitivity of the sensor along its length. An overview of the typical configurations of WIM systems equipped with piezoelectric, load cell, and bending plate sensors, respectively, was described in [25]. However, the equipment of these systems was limited only to load sensors and induction loops. The concept of identifying the traffic structure of the vehicles crossing the bridge, combined with information on the size of the transported load, we can find in [26]. The WIM system containing a single line of load sensors was additionally equipped with three cameras. The class of the weighed vehicles is identified based on the analysis of the images from these cameras. Sometimes, new sensor designs intended for WIM systems are presented. In [27] is analysed a force sensor that consists of two spring washers welded to a thin ring in the middle and two strain gauges. As a result of compressing the sensor, the ring with strain gauges becomes tense. The sensor allows for the measurement of three force components. The presented WIM system is equipped with a single line of such sensors. The justification for the need to test the properties of WIM systems operating in various operating conditions is the data contained in [28]. The authors of this report identified the root causes of the problems with the data from WIM systems observed by their users. Users mentioned problems with load sensors mostly (78%). Also, the change in the accuracy of the weighing results during the operation of the WIM system was mentioned (62%). The WIM system described in our work allows for long-term research that can explain the causes of these problems and develop solutions to prevent the observed dysfunction of WIM systems. A test WIM system with a similar purpose to that described in this paper was presented in [29]. It has 12 load quartz sensors, 10 inductive loops, 6 position sensors, 7 temperature sensors at various depths, and 9 geophones and accelerometers. However, the system is not equipped with rainfall, wind, icing, or humidity sensors, nor with cameras enabling the recognition of the vehicle’s registration number and its unambiguous identification. This identification is important when the vehicle is re-weighed for verification on a low-speed scale with high precision. A WIM system with a relatively simple sensor configuration is presented in [30]. The described WIM system included two lines of strain gauge load sensors, two induction loops, and an overview camera enabling the observation of vehicle movement at the WIM station. Additionally, an ANPR camera was installed at the WIM station, which allows for the reading of the vehicle’s registration number. However, the system is not equipped with any meteorological parameter sensors.

The WIM system described in this paper has original features that enable the conducting of innovative research. Such research is necessary and particularly important in the context of using WIM technology in the direct mass enforcement systems of commercial vehicles. The special features of the built WIM system are as follows:The system is equipped with eight lines of load sensors made using two different technologies, i.e., four lines of quartz sensors and four lines of bending plate sensors. Both types of sensors are installed at a short distance from each other, which ensures that they work in identical conditions determined by the material and pavement parameters, road traffic conditions such as traffic intensity, traffic structure, and speed distribution, and environmental parameters,The system is equipped with temperature sensors, placed in the immediate vicinity of each load sensor, and additionally with a temperature sensor enabling the measurement of the temperature distribution deep into the road substructure, up to a depth of 30 cm, with a resolution of 5 cm,The system is equipped with sensors for environmental parameters such as wind direction and intensity, the occurrence and intensity of rainfall, the presence of a water layer or snow on the pavement, air humidity, and temperature,The system is equipped with cameras enabling the observation of the vehicle trajectory at the weighing station and cameras enabling the recognition of vehicle plate numbers. This allows for the direct control of the weighing accuracy by directing the selected vehicle (based on the plate number) to a low-speed scale, built in the immediate vicinity. The low-speed scale has a metrological verification certificate.

This configuration of the described WIM system not only enables the testing of the impact of all the above-mentioned factors on weighing accuracy. Additionally, it is possible to compare the sensitivity of both load sensor technologies to these factors. The built system also makes it possible to analyse the impact of the number of load sensors on the weighing accuracy and to study the impact of various data fusion algorithms on this accuracy. The weighing accuracy of each vehicle can be directly verified at a low-speed scale.

Additionally, this configuration of the WIM system enables the testing of various strategies and algorithms for verifying the reliability of the weighing results. The reliability of the weighing results is particularly important in direct mass enforcement systems. The high reliability of the results allows us to minimize the probability of classifying a standard vehicle as overloaded.

The authors are not aware of WIM systems with a similar configuration that have such metrological instrumentation and thus enable such comprehensive tests.

## 3. Structure of the WIM System

The WIM system is located on the DW 975 Provincial Road in southern Poland. It is equipped with two sets of load sensors. The first set consists of eight quartz sensors from Kistler, arranged in four lines (SA, SB, SC, SD). The second set includes eight bending plate sensors from PAT, also arranged in four lines (SE, SF, SG, SH), each 3.5 m long. This design of the load sensors makes it possible to measure the load of each wheel of the vehicle four times, on each set of sensors. Inductive loops and piezoelectric sensors have been installed between the load sensors to control the correct passage of the vehicle. The scheme of the constructed station is shown in Figure 2. In Figure 3, a part of the investigated WIM system equipped with bending plate sensors is presented.

The built-in multi-sensor HS-WIM system is additionally equipped with a set of sensors that make it possible to measure environmental factors, enabling the study of their impact on the results of the weighing of the vehicles. The installed sensors allow for the measurement of the strength and direction of the wind, detection of precipitation, moisture on the pavement, flooding of the surface with water, and salinity of the pavement. In addition, the set of temperature sensors measures the temperature in the substructure of the road to a depth of 30 cm, with a spatial resolution of 5 cm, the pavement temperature at 0 cm and −5 cm at the ends of the load sensors, as well as the air temperature. Two cameras allow for the monitoring of the road traffic at the WIM station and recognition of the plate numbers of weighed vehicles.

In the immediate vicinity of the WIM station, a low-speed scale was constructed (Figure 4). The scale has a metrological legalisation certificate and is used to verify the accuracy of the weighing result at the WIM station. Thanks to the cooperation with the Inspectorate of Road Transport, which is the institution authorised to carry out the load control of motor vehicles moving on public roads in Poland, there is access to the results of weighing vehicles on a legalised scale, including those vehicles that were previously weighed at the WIM station. The vehicle is identified by its registration number.

## 4. Assessment of Pavement Quality

A very important issue from the point of view of the accuracy of the WIM system is the quality of the pavement, because, as an excitation for a moving vehicle, it causes the appearance of a dynamic component in the signal of the load of its wheels on the road’s surface. The parameters describing this quality include the curvature of the road; longitudinal and transverse slopes, usually measured by the geodetic method; road irregularities in the range of low and high spatial frequency values, measured by profilometers equipped with laser devices; and the deflections of the pavement under static or dynamic conditions (in this case, one of commercially available deflectometers is usually used).

In order to determine these parameters, geodetic surveys were carried out to determine the straightness of the road at the WIM site; measurements were made of the longitudinal section of the road and the longitudinal and transverse inclination. The straightness of the road section with the installed WIM system was checked using an electronic tachymeter from Leica (Figure 5) which operates in the infrared spectrum, at a wavelength of 780 nm, at a distance range of 1.5–300 m [31]. The measurement was carried out over a 230 m long section covering 150 m in front of the first WIM station sensor, 30 m along the WIM station, and 50 m behind the last load sensor. Based on the results obtained, it was determined that this section of the road is straight (the radius of the curvature is large enough).

The height measurement of the longitudinal profile on a section of 230 m was carried out using the precision levelling method and inverse patches (the measurement accuracy of leveller is 0.3 mm) [32]. The position of the measurement points (located approximately 10 cm from each other) was determined using a satellite receiver. The total accuracy of the determination of the height of the measuring points on the profiles, due to the properties of the tested pavement and the selected measurement method, is approximately ±1 mm. The longitudinal profile measurements were performed with a spatial resolution of 10 cm. The measurements of the transverse profile were made in four sections of the road, i.e., on the first and last quartz load sensor and on the first and last bending plate sensor. The obtained geodetic characteristics of the road section in the vicinity of the WIM site are shown in Figure 6 and Figure 7.

The obtained measurement results indicate that the cross-section meets the requirements formulated for WIM stations [3]. The lateral slope of the pavement is approx. 2% towards the side (sign as zero on *x*-axis), which is the result of the treatment used for the efficient drainage of rainwater. The longitudinal section of the road on the section approx. 100 m in front of load sensors and approx. 20 m behind the sensors is practically linear, and the difference in levels at a distance of approximately 150 m is less than 100 cm, which gives an inclination of less than 0.7%. On the section where the load sensors are installed, the difference in levels is 20 cm, which indicates the same slope as on the entire length of the tested section.

In addition to the longitudinal and transverse slope of the road, the road roughness of the pavement determines the quality of the WIM systems. The quality of the pavement in which the WIM load sensors are installed is one of the basic factors affecting the accuracy of weighing vehicles in motion. Smooth pavements, without significant changes in profile height, prevent the significant vertical bouncing of the weighed vehicle during its passage through the measuring station. Such pavements are considered to be of high quality; WIM stations installed in such pavements provide a higher weighing accuracy. When assessing the pavement, both slow changes in the profile height, observed over a significant length of the road, reaching several tens of metres, as well as fluctuations in the profile height observed over a length of several tens of centimetres, are taken into account. These changes in the height of the road profile are characterised either by the wavelength expressed in metres or by the spatial frequency of changes in the height of the profile expressed in the number of cycles per metre of road length.

To assess the quality of the pavement in the range of short waves, a laser profilometer (which directly measures the current distance from the measuring beam to the road surface point) with three beams scanning the surface was used. Next is calculated the roughness of the pavement in the range of a higher frequency. A vehicle with a mounted scanner (Figure 8) drove through a section of road with a length of approx. 500 m (350 m in front of the WIM station and 150 m behind) nine times. The transverse track position of each lane crossing was randomly selected to scan the width of the entire lane uniformly. The result was scans of 27 tracks, sampled every 10 cm along the road. To facilitate the location of the WIM sensors on the scans from the selected road section, a reflective tape was pasted on the first and last load sensor in the system (Figure 9).

Sample scans of three tracks obtained during a single crossing are shown in Figure 10. They represent the result of measuring the height of the pavement profile as a function of the track travelled by the scanner.

The assessment of the pavement was carried out in accordance with the ISO 8608 [33]. The basis for this assessment is the displacement power spectral density (PSD) of the road profile, expressed as a function of the spatial frequency, denoted by *n*. Spatial frequency *n* it is expressed in units [number of cycles/m]. The frequency resolution of this characteristic is determined by the length of the scanned road section. In the present case, it is Be=1500 m=0.002 [cycles·m−1]. The accepted coordinate system in which this characteristic is presented is defined in ISO 8608.

The ISO 8608 standard distinguishes eight classes of pavement quality marked with the letters A—H. The upper limit value of PSD, specified as a spatial frequency of less than 0.1 $\mathrm{cycles}\cdot m^{-1}$, is PSD = 32 ∙ 10^−6^ m^3^ for the best class road marked as A, and for the worst class marked as H, the permissible PSD value may be greater than 0.131 m^3^. The characteristics of the PSD in a semi-logarithmic coordinate system are represented of straight lines constituting the boundaries between the successive classes of pavements, which greatly facilitates the assessment of the class of the tested pavement. It is represented both as a function of spatial frequency *n* as well as in the angular spatial frequency *Ω*, expressed in [rad∙m^−1^]. Both frequencies are linked by a simple relationship: *Ω* = 2π∙*n*.

Figure 11 shows the PSD relationship (denoted as G_d_) from frequencies *n* and *Ω*, determined on the basis of a single scan of the road profile (one of 27 recorded during the tests). The wavelength λ also shown in this figure is related to the spatial frequency by a known relationship 1/n. The same figure shows the frequency-smoothed PSDs. The smoothing was carried out in the frequency ranges specified in ISO 8608. In addition, the drawings show a straight line approximating the specified PSD characteristics (red).

Based on the results obtained, it can be concluded that the pavement in which the WIM station was installed belongs to Class A, and, therefore, is the best possible one. An analysis of the profiles of the other registered profiles leads to a similar conclusion.

Based on the recorded data from the profilometer, the IRI (International Roughness Index) indicator, which characterises the quality of the pavement, was also estimated. The IRI is expressed in mm/m or m/km. It is an international indicator of pavement roughness. It characterises the operation of the suspension in the generally accepted calculation model of a vehicle which moves at a constant speed of 80 km/h, on a recorded profile of the pavement, on a section of road with a length of 50 m [34]. For a Class A road, the average value of the indicator may not exceed 1.3 mm/m and the maximum value may not exceed 2.4 mm/m. For the test section on which the described WIM system was installed, in the worst case, i.e., at the installation site of the sensors (interference with the pavement), the results of 1.1 mm/m and 2.0 mm/m IRI were obtained, respectively, which confirms the results obtained on the basis of the spectral analysis.

The dynamic deflection of the pavement was also tested with an FWD device. The measurement was carried out at five measuring points, i.e., 5 and 15 m in front of the first sensor of the system, in the middle between the quartz and bending plate sensors, and 5 and 15 m behind the last sensor of the WIM system. The pavement was loaded using “falling weight” with a force of 50 kN over 20 ms, which corresponds to a vehicle traveling at a speed of about 60 km/h. The deflections were measured with geophones at seven points 0–1800 mm from the point where force was applied, with an interval of 300 mm. The method allows for the deflection to be measured with an accuracy of 2.5%. The measurements were taken at a pavement temperature of 13 °C. At no point did the pavement deflection exceed 223 µm. All measurement results are in the range of 150–223 µm, which makes it possible to classify the pavement as “class II”, according to [3]. The process of measuring the deflection of the pavement at the WIM test station is shown in Figure 12.

## 5. Exemplary Measurement Results from the WIM Station

The WIM station was launched in July 2021, and since then, the measurement data has been collected on a dedicated server. The initial period of the operation of the station, i.e., from July to the end of 2021, was intended for the testing and tuning of the measurement systems, elimination of potential defects, verification of recorded measurement data, agreement of data formats collected on the server, and preliminary calibration of weighing systems. The proper operation of the systems, and thus the registration of the measurement data, began in January 2022 and continues to this day. The number of results of the weighing of vehicles at the station depends on the traffic volume in a given period and is within the range of 100,000 results vehicles/month in winter, with up to 140,000 vehicles/month during the summer.

The data is aggregated for the period of each month, i.e., data from the full calendar month, and is stored in a single file for each weighing subsystem separately. Thus, three data files are created each month: one METEO (with data from the weather station) and two for the WIM1 and WIM2 vehicle weighing subsystems. A total of 583 variables are recorded. In each subset of data, the first variable is the measurement time. This allows for the synchronisation of data from different sources (METEO, WIM1, and WIM2), and the second parameter allowing for the proper combining of the relevant records is the vehicle registration number.

### 5.1. Meteorological Data

Below, we present exemplary characteristics illustrating the variability of the selected environmental quantities during a selected month. The presented results were registered in September 2022.

#### 5.1.1. Pavement Temperature Measurement

There are a total of 16 temperature measuring points located at the axle load sensor lines (two at each sensor). Figure 13 shows the results of measuring the pavement temperature, from a measuring point located at the third line of the quartz load sensors. At each measuring point, the temperature is measured just below the pavement surface (0 cm), and 5 cm below the pavement (−5 cm, which corresponds to the depth of the location of the sensitive load sensor element).

The dynamics of the temperature change measured at the depth of the installation of the load sensors is slightly less than at the surface. A more in-depth analysis of the temperature variation throughout the weighing station enables the estimation of possible temperature gradients along and across the station. The possible occurrence of such a gradient is relevant for the analysis of the accuracy of the weighing results.

#### 5.1.2. Road Substructure Temperature Measurement

In addition to the temperature sensors located along the load sensor lines, a temperature sensor was installed at one point of the WIM station deep in the road substructure. It is designed to measure temperature at depths of 0, −10, −15, −20, −25, and −30 cm below the road surface. Figure 14 shows the measurement results from this sensor.

The presented results illustrate the temperature gradient in the depth of the road substructure. As expected, the dynamics of the temperature changes are less the deeper the sensor is located. In summer, the temperature at greater depths is lower than at the surface (August). In the autumn–winter period, the stabilisation of the temperature distribution can be observed (November).

#### 5.1.3. Measurement of Parameters Determining the Condition of the Pavement

Another group of sensors installed on the station is used to measure a number of quantities associated with the presence of a layer of water on the pavement. On this basis, it can be assessed not only whether there has been precipitation at the station, but also whether the pavement is damp, for example due to the condensation of water vapour, or is flooded with a layer of water after heavy precipitation. The measured parameters are defined as follows:“Pavement condition”—this variable takes quantised values from zero to five, with subsequent values indicating the following: 0—dry surface, 1—damp, 2—wet, 3, 4—increasing degrees of wetness, 5—flooded (running/standing water),“Precipitation state”: 0—no precipitation, a value equal to or greater than 1 indicates precipitation, and a numerical value indicates the percentage of precipitation time,“Water layer”—the thickness of the water layer at the installation site of the sensor, expressed in mm.

Sample results are shown in Figure 15. In the course of the planned further studies, a correlation will be sought between the accuracy of the weighing result and the parameters characterising the condition of the pavement.

#### 5.1.4. Wind Speed and Direction Measurement

In addition to the dimensions associated with the condition of the pavement, the speed and direction of the wind are also measured, which can also affect the accuracy of the weighing results of the vehicles. Figure 16 shows the sample results obtained in September 2022. The direction of 0 degrees corresponds to north. In addition to the above-mentioned meteorological values, the system also records or calculates based on measurements such values as air temperature, perceptible temperature, dew point temperature, atmospheric pressure, and air humidity. However, from the point of view of assessing the metrological properties of WIM systems, it seems that these are of secondary importance.

## 6. Assessment of System Accuracy

The accuracy of both WIM systems was assessed using the pre-weighed vehicle method. During the experiment, three vehicles were used: a two-axle vehicle (rigid body), a three-axle vehicle and a five-axle vehicle (a two-axle tractor with a three-axle semi-trailer). These vehicles were weighed on a platform scale of accuracy class 0.2 and on a low-speed scale of accuracy class D2. The reference value for the gross vehicle weight of each reference vehicle (GVW) was the result of its weighing on the platform scale. The reference value for single and multiple axle load is the adjusted mean of five multiple/single axle load measurements carried out using the LS-WIM scale constructed in the vicinity of the described WIM system. The determination of the corrected load value and the calculations leading to the determination of the reference values were carried out in accordance with the recommendations [17]. The data are summarised in Table 1.

The GVW and reference values of axle load values determined for all three vehicles were the basis for the error estimation of both WIM stations. The test vehicles drove through both WIM stations 15 times at a speed of approx. 50 km/h, 60 km/h, and 80 km/h, making five runs at each speed.

The distribution of the GVW measurement errors at a WIM station equipped with bending plate sensors is shown in Figure 17. The reference value was the result of weighing the vehicles on the platform scale. When analysing the presented results, it should be taken into account that the assessment of the accuracy of both WIM stations was carried out after a period of almost two years of their operation, without the prior correction of the calibration factors. According to the purpose of the system for direct mass enforcement, maximum error values are considered.

From the characteristics shown in Figure 17, the following conclusions can be drawn:The smallest range of weighing errors, based on the measurements from a single load sensor, occurs in the case of a five-axle vehicle. The maximal value of error in this case is approximately 8% for all vehicles,By averaging the weighing results obtained on each of the four load sensors, the GVW estimation error was reduced approximately twofold. After averaging, the measurement error of the GVW of the two-axle vehicle did not exceed 3.5%, and for the three-axle and five-axle vehicles, 5%.

The experiment also made it possible to evaluate the errors in estimating the static load of a single axle. The results obtained allow us to formulate the following conclusion:The static load measurement error of a single axle of a two-axle vehicle on a single load sensor is approx. 10%,For three-axle and five-axle vehicles, this error is approx. 15%,Averaging the weighing results on the four load sensors reduced this error to 6% for a two-axle vehicle, approx. 8% for a three-axle vehicle and approx. 10% for a five-axle vehicle.

Similar tests were carried out for the WIM system equipped with quartz load sensors. The distribution of the GVW measurement errors are shown in Figure 18.

The results obtained allow us to draw the following conclusions about the accuracy of weighing:The gross vehicle weight (GVW) of the vehicles is determined with a relative error not exceeding 6% for each pre-weighed vehicle (two-axle vehicle, three-axle vehicle, and five-axle vehicle, respectively) when weighing on a single load sensor,Averaging the weighing results on the four load sensors reduces this error to 3.5% for a two-axle vehicle, 4% for a three-axle and 4.5% for a five-axle vehicle,The static load measurement error of a single axle, on a single quartz load sensor, is 6% for a two-axle vehicle, 8% for a three-axle vehicle and 12% for a five-axle vehicle,Averaging the axle load measurements from each of the four quartz load sensors reduces this error to approx. 6% for two-axle and three-axle vehicles and approx. 8% for five-axle vehicles.

The error distributions shown in Figure 17 and Figure 18 support the claim that both WIM stations (one equipped with four strain gauge bending plate load sensors and the other equipped with four quartz load sensors) provide comparable weighing accuracy. Given that the results were obtained after a two-year period of operation, such a result should be considered satisfactory. The change in the calibration coefficients of both WIM stations, which can be carried out based on the collected measurement results, should allow for a significant reduction in weighing errors, especially since the determined error distributions are generally not symmetrical.

## 7. Summary

In the community of road administrators, there is a common conviction of the need to introduce the continuous and universal control of the weight of motor vehicles. The procedures for the type approval and metrological testing of WIM systems used in direct mass enforcement should ensure that the likelihood of a wrongful penalty being imposed on the carrier or loader of a vehicle that is in fact normative and has simply been inaccurately weighed is minimised. It is therefore necessary to test the assessment procedures and the assessment rules and requirements for such systems. Their effectiveness must be tested in real traffic conditions. Work aimed at achieving this goal is also being carried out in Poland. The described WIM station serves just such a purpose. Its structure, equipment with very wide measurement capabilities, and installation on the road on which the actual movement of vehicles takes place mean that all previously formulated requirements and conditions have been met. The results of the measurements collected during the twelve months of its operation fully confirmed the achievement of the accepted assumptions. The system is fully functional, allowing for the measurement of many parameters of both the vehicle being weighed as well as parameters potentially affecting the accuracy of the weighing. The assessment of the accuracy of the system indicates the stability of its metrological parameters. The equipment of the system with sensors of environmental parameters allows for a search for correlations between the accuracy of weighing and the intensity of these parameters, in the range observed in the climatic conditions prevailing in Poland, and to introduce a possible correction of the results or determine maps of the accuracy of the system [35]. Extreme weather phenomena do not occur in Poland, and if they do, they are episodic. In summer, the maximum air temperature rarely exceeds 35 °C, and these extreme values occur in short, several-day periods. In winter, the air temperature generally does not drop below several degrees Celsius. The permissible speed of motor vehicles outside built-up areas is 90 km/h, on expressways is 110 km/h, and on highways is 140 km/h. The permissible GVW of five-axle vehicles is 40 tons. All these factors cause certain limitations in the scope of research on the WIM system that can be carried out.

A summary of the weighing errors determined for the described WIM system is presented in Table 2.

The weighing accuracy in the system with the load sensors of both technologies is comparable. The maximum error of the GVW measurement on a single load sensor is 6% for quartz sensors and 8% for bending plate sensors. The value of this error does not depend on the vehicle class (two-axle vehicle, three-axle vehicle, five-axle vehicle). Averaging the weighing results from four load sensors allowed us to reduce the maximum error to 4.5% for quartz sensors and 5% for bending plate.

For a single quartz sensor, the maximum error in measuring the static load of a single axle ranges from 6% for two-axle vehicles to 15% for five-axle vehicles. In the case of the bending plate sensor, this error ranges from 10% for two-axle vehicles to 15% for five-axle vehicles. Averaging the results from four quartz load sensors allowed us to reduce the maximum error to a value from 6% for two-axle vehicles to 12% for five-axle vehicles. For bending plates, these values range from 6% to 10%, respectively.

The aim of the research planned to be carried out at the described WIM station is to determine the impact of basic parameters on the weighing accuracy. The parameters taken into account in the planned tests are the speed of the weighed vehicle, road temperature, wind speed and direction in correlation with the direction of travel of the weighed vehicle, humidity, rainfall and its intensity, and icing. Environmental sensors were installed in the road surface (temperature, humidity, rainfall, icing sensors) and outside the traffic lane, in the immediate vicinity of the WIM station (wind speed and direction, air temperature, the amount of rain). In particular, 17 road temperature sensors were installed at the station, two sensors for each line of quartz load sensors and 9 temperature sensors for bending plate load sensors. These sensors enable temperature measurement at a depth of −5 cm, which corresponds to the depth at which the sensitive elements of the load sensors are located. This is justified by the fact that the pavement temperature may differ significantly from the temperature measured at different depths. It may also change quickly under the influence of sunlight or, for example, rainfall. However, due to thermal inertia, rapid and short-term changes in pavement temperature do not cause significant temperature changes at greater depths. The WIM station is also equipped with an integrated sensor enabling temperature measurement to a depth of −30 cm, with a spatial resolution of 5 cm.

The planned research will result in characteristics illustrating the influence of the mentioned parameters on the vehicle weighing error. These characteristics will be determined within the ranges of the variability of influential parameters, typical for the climatic conditions prevailing in Poland.

## Figures and Tables

**Figure 1 sensors-23-09290-f001:**
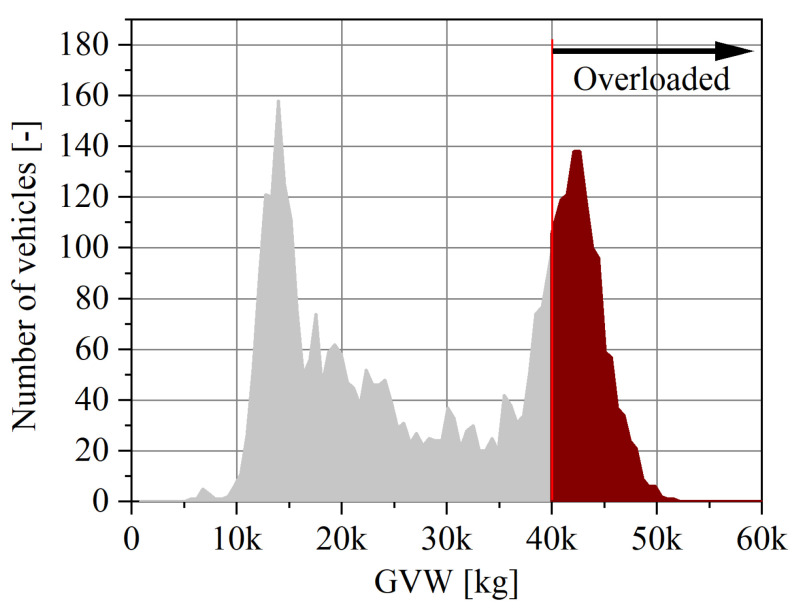
Histogram of the distribution of gross weight of 5-axle vehicles (2-axle tractor + 3-axle semi-trailer.

**Figure 2 sensors-23-09290-f002:**
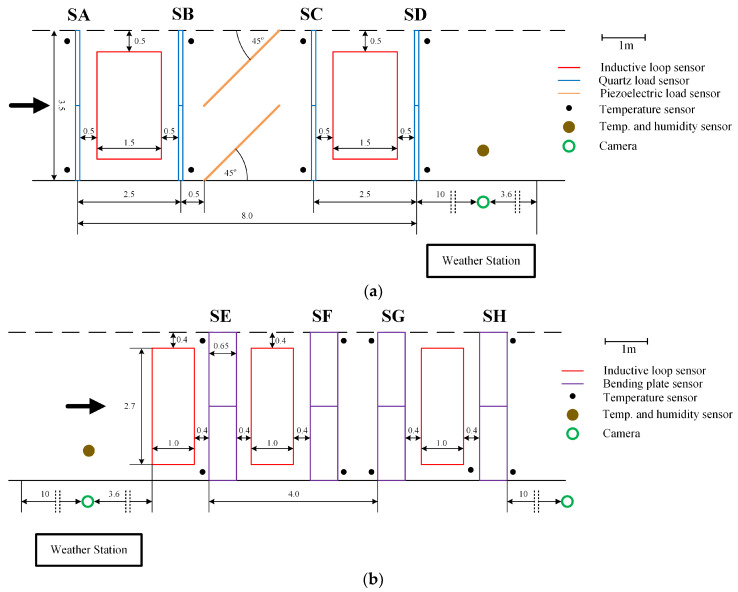
Structure diagram of the designed WIM site: (**a**) the section with quartz sensors (WIM1); (**b**) the section with bending plate sensors (WIM2).

**Figure 3 sensors-23-09290-f003:**
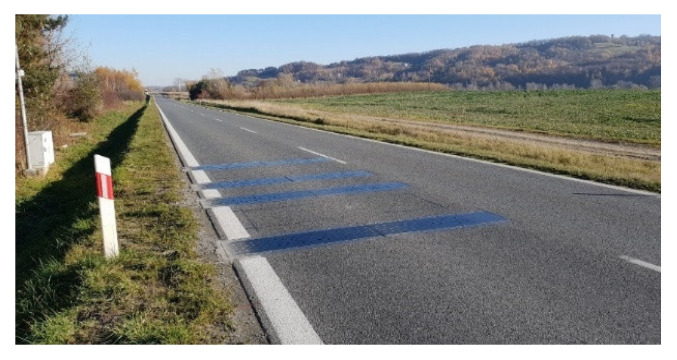
A fragment of a WIM system with strain gauge bending plate sensors.

**Figure 4 sensors-23-09290-f004:**
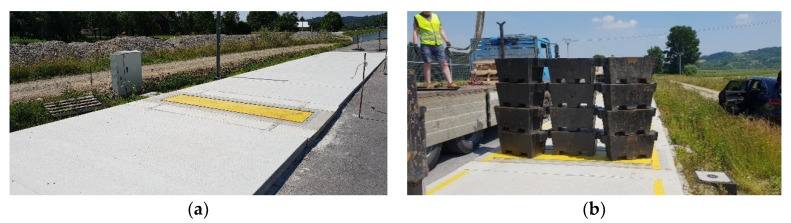
LS-WIM low-speed scale (**a**) and an illustration of the procedure for its initial legalisation (12-ton load) (**b**).

**Figure 5 sensors-23-09290-f005:**
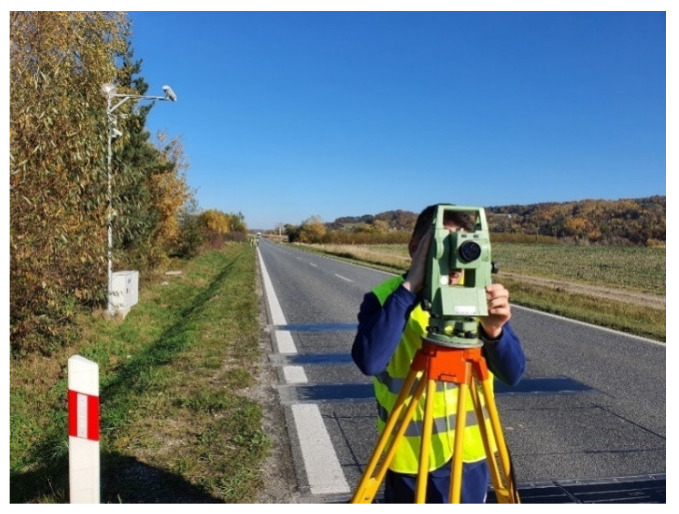
Geodetic measurements at the WIM station.

**Figure 6 sensors-23-09290-f006:**
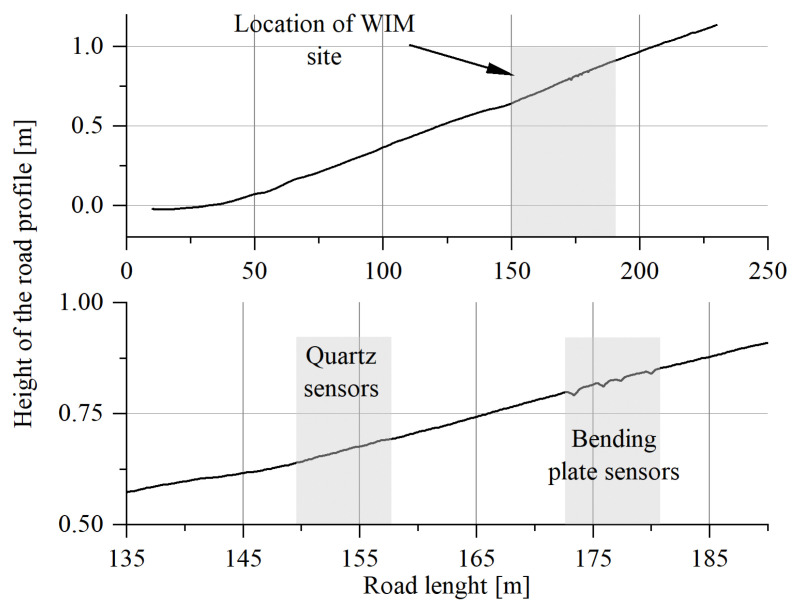
The longitudinal section of the road profile (upper) and the longitudinal section at the load sensor installation site (lower). Quartz sensors from Kistler (SA, SB, SC, SD); Bending plate sensors from PAT (SE, SF, SG, SH).

**Figure 7 sensors-23-09290-f007:**
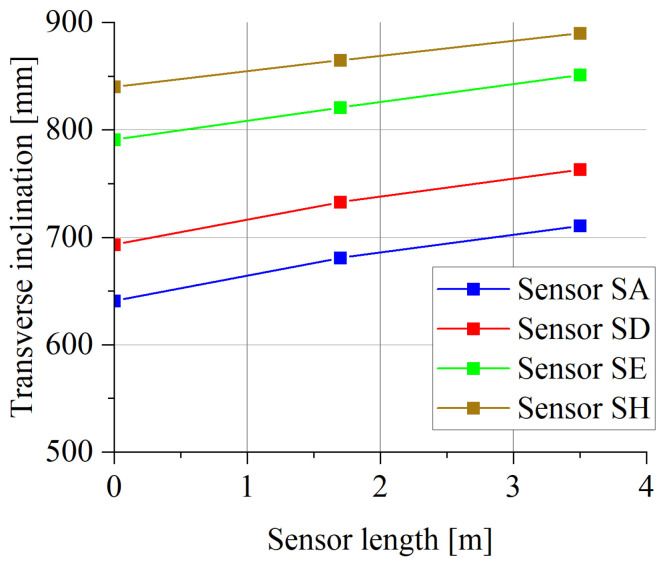
Cross-section of the road in four selected points.

**Figure 8 sensors-23-09290-f008:**
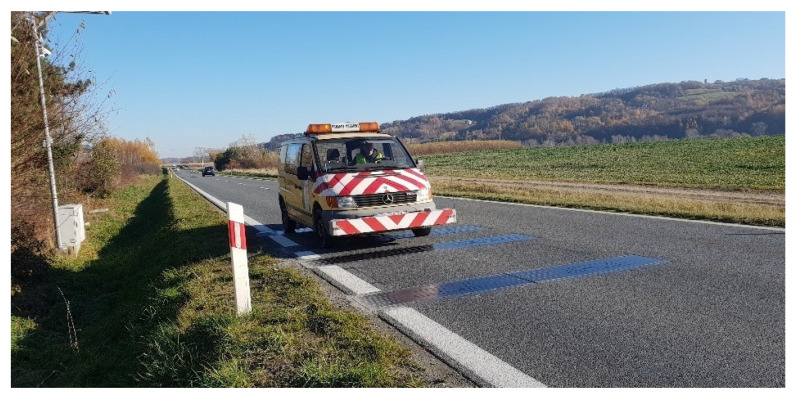
The passage of the vehicle with the profilometer through the WIM station.

**Figure 9 sensors-23-09290-f009:**
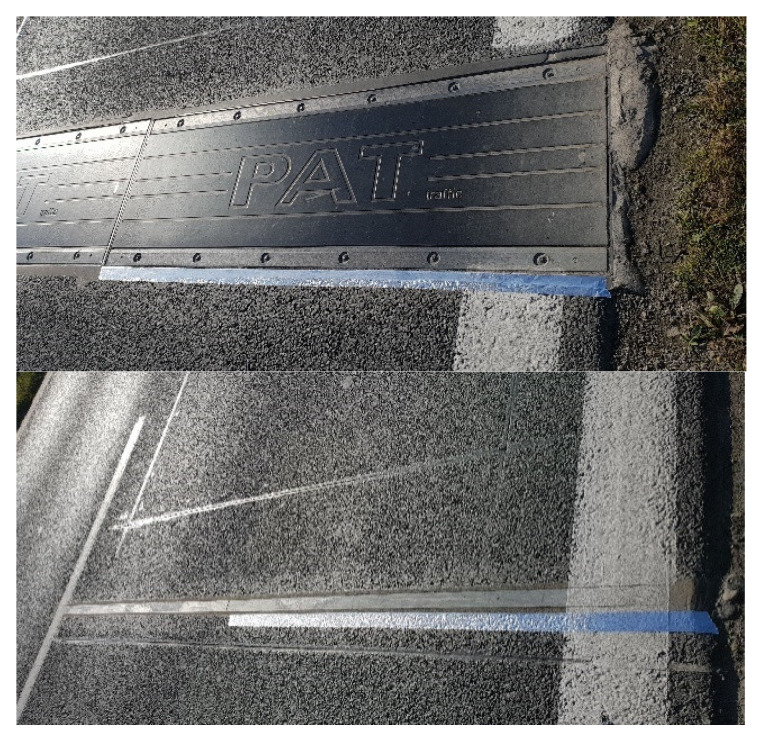
Reflective strips marking the position of the beginning and end of the road section on which bending plate and quartz load sensors are installed.

**Figure 10 sensors-23-09290-f010:**
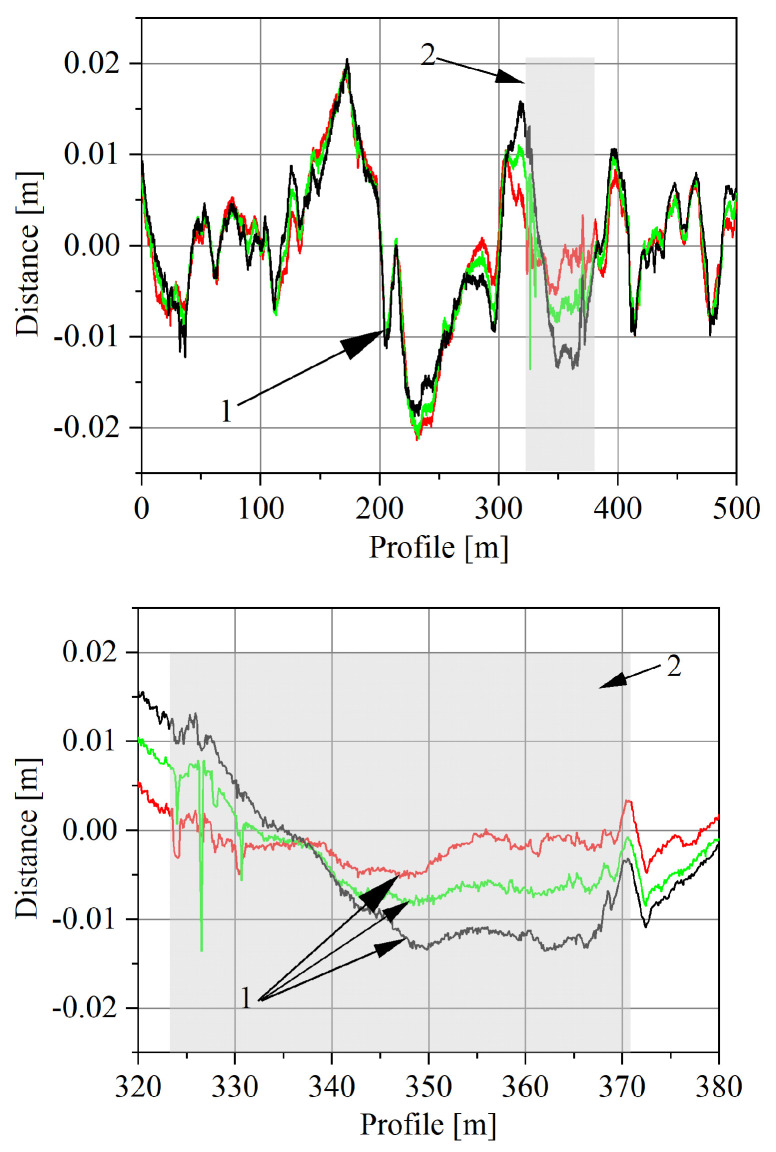
The course of the pavement profile as a function of the road. 1—The result of registration of three tracks during one pass, 2—the boundaries of the section on which the load sensors are installed.

**Figure 11 sensors-23-09290-f011:**
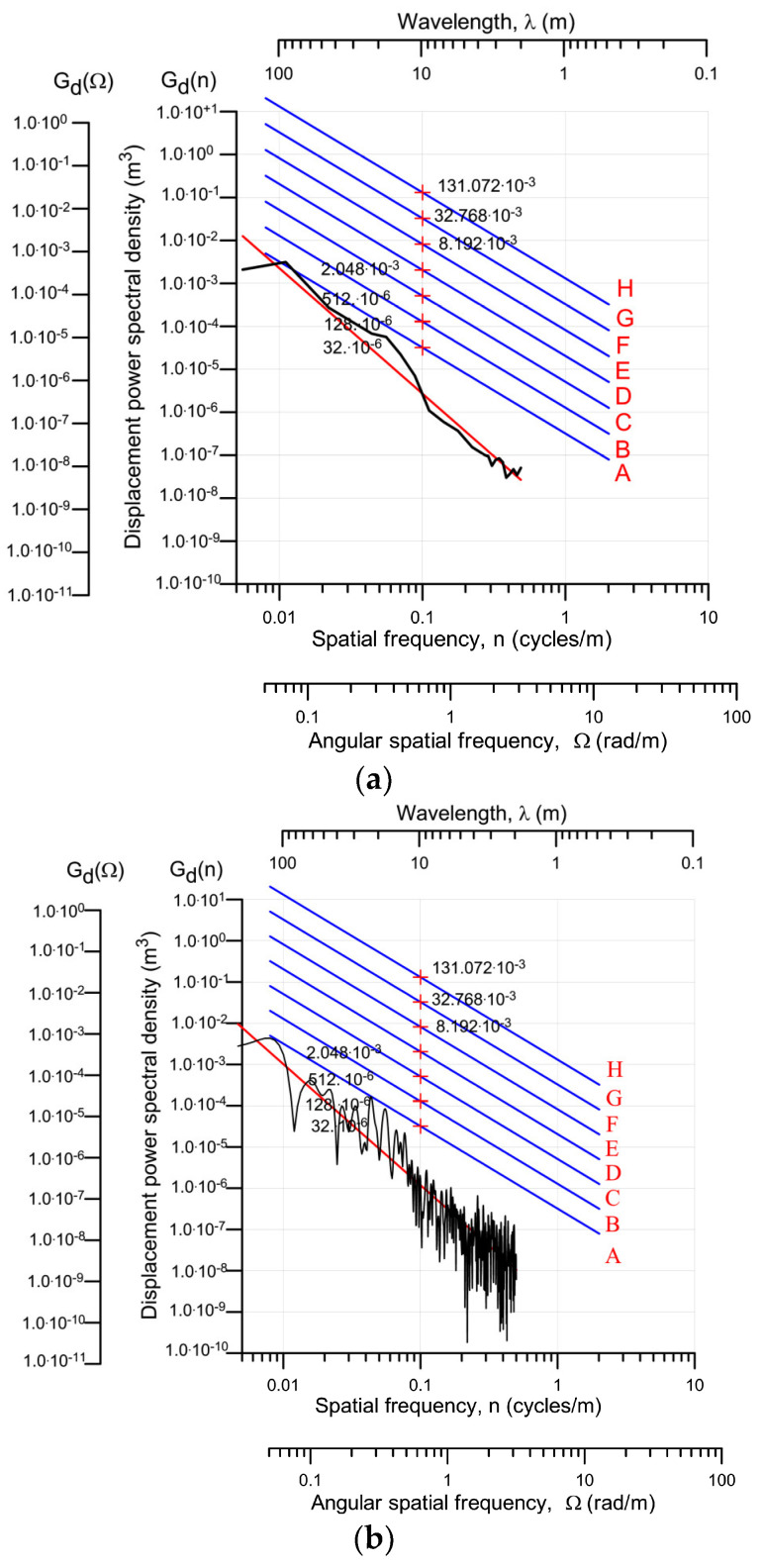
The power spectral density (PSD) of the road profile for a single scan track. (**a**)—Characteristics averaged over successive frequency ranges (in accordance with the standard), (**b**)—characteristics determined on the basis of a single scan track without smoothing. A–H—classes of pavement quality (A—is the best class).

**Figure 12 sensors-23-09290-f012:**
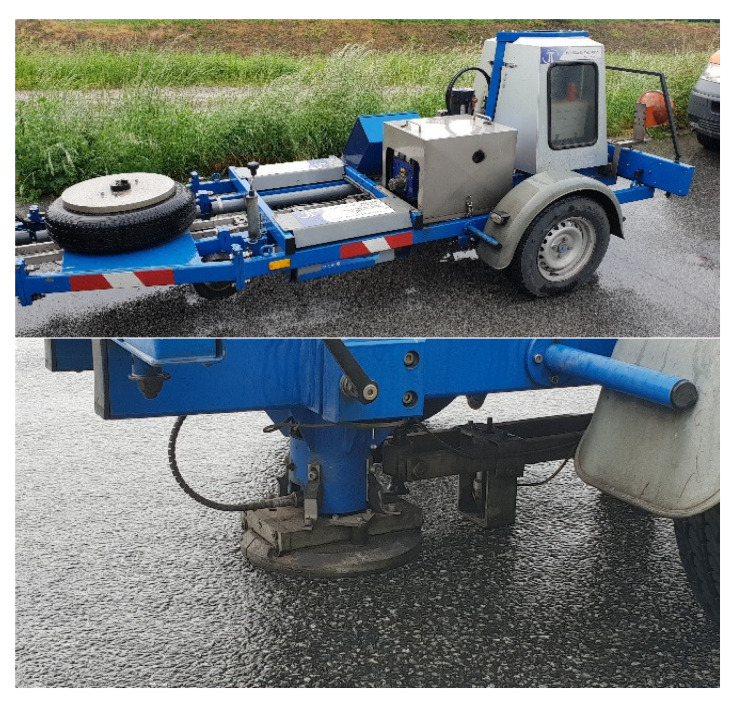
Pavement deflection measurement using FWD.

**Figure 13 sensors-23-09290-f013:**
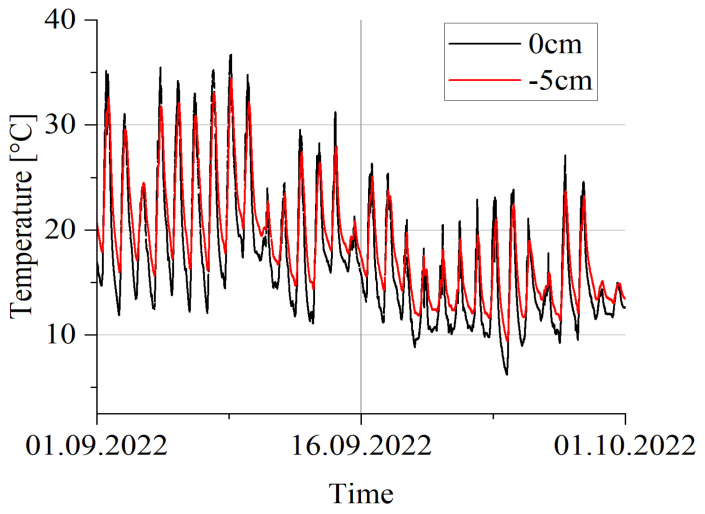
The results of the measurement of the temperature on the road surface and the depth of −5 cm.

**Figure 14 sensors-23-09290-f014:**
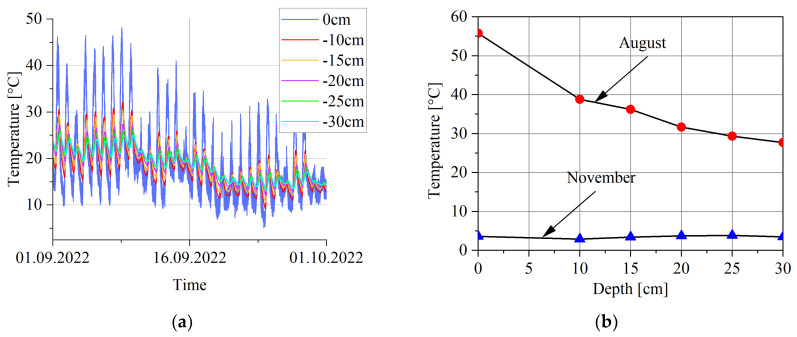
The results of measuring the temperature of the road substructure at depths from 0 to −30 cm (**a**) and the dependence of temperature on depth (**b**).

**Figure 15 sensors-23-09290-f015:**
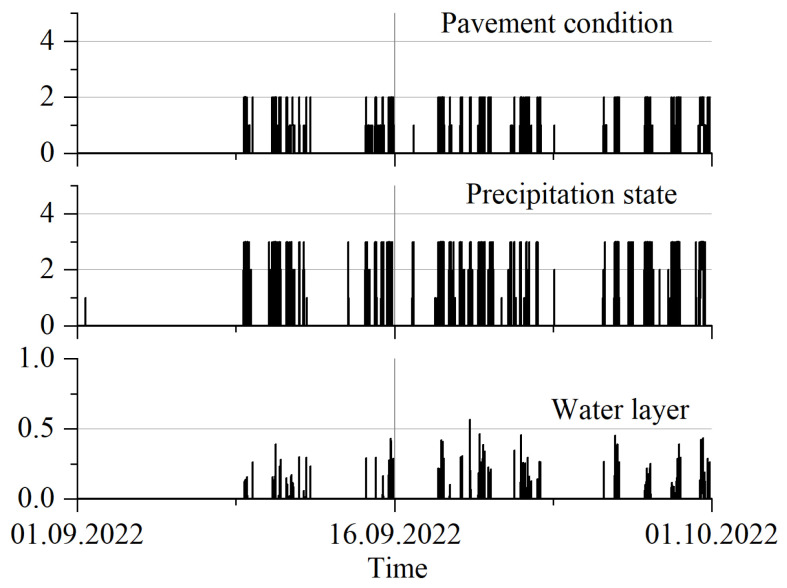
Measurement results related to the current condition of the pavement.

**Figure 16 sensors-23-09290-f016:**
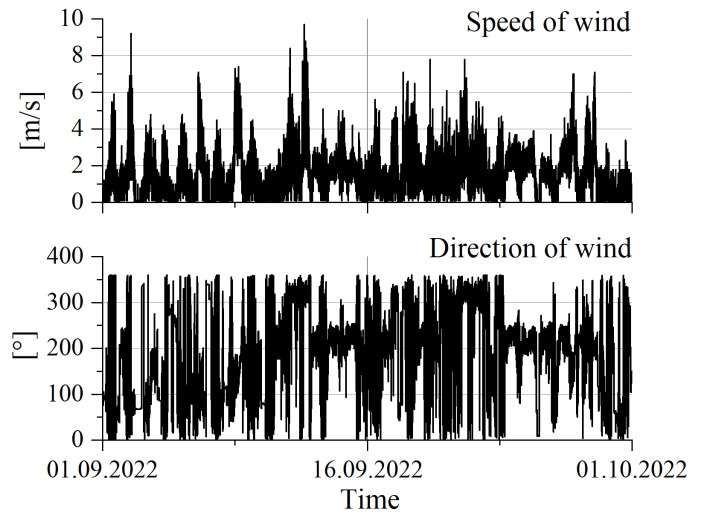
Measurement results related to the current wind speed and direction at the measuring station.

**Figure 17 sensors-23-09290-f017:**
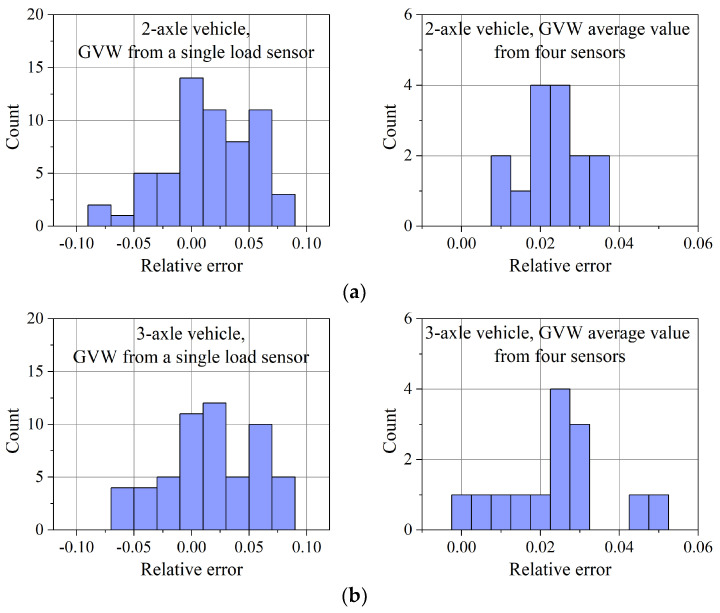
Results for a WIM station equipped with bending plate load sensors. Distribution of GVW estimation errors based on the weighing result on one exemplary load sensor and as an average of the results from four load sensors for: (**a**) 2-axle vehicle, (**b**) 3-axle vehicle, and (**c**) 5-axle vehicle.

**Figure 18 sensors-23-09290-f018:**
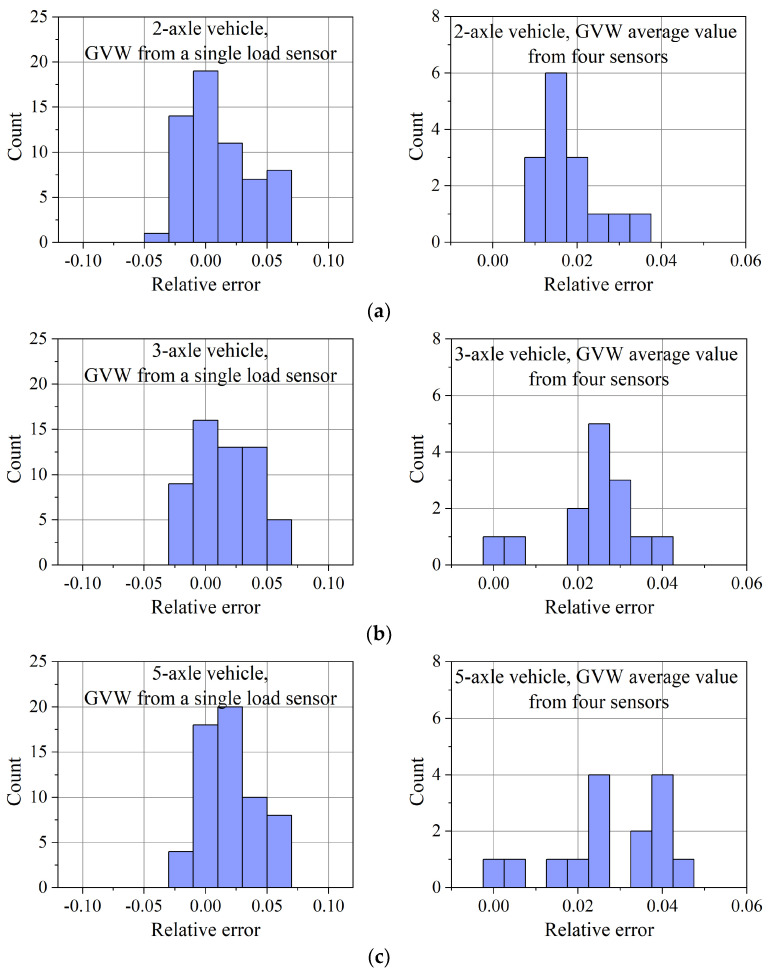
Results for WIM system equipped with quartz load sensors. Distribution of GVW estimation errors based on the weighing result on one exemplary load sensor and as an average of the results from four load sensors for: (**a**) 2-axle vehicle, (**b**) 3-axle vehicle, and (**c**) 5-axle vehicle.

**Table 1 sensors-23-09290-t001:** The results of weighing of the reference vehicles on the platform scale and on the LS-WIM scale.

	GVW [kg]	Axle Load I			Axle Load II			Multiple Axle Load		
		Mean Value [kg]	Ref. Value [kg]	Std [kg]	Mean Value [kg]	Ref. Value [kg]	Std [kg]	Mean Value [kg]	Ref. Value [kg]	Std [kg]
2-axle vehicle	16,240	6296	6298	26.1	9940	3855	49	x	x	x
3-axle vehicle	25,380	7268	7268	22.8	x	x	x	18,112	18,112	54.0
5-axle vehicle	39,860	7160	7184	14.1	11,308	11,346	10.9	21,260	21,331	31.6

**Table 2 sensors-23-09290-t002:** Summary of weighing errors determined for the described WIM.

	GVW—Maximal Error [%]						Static Load of Single Axle—Maximal Error [%]					
	From Single Sensor			Average Value from Four Sensors			From Single Sensor			Average Value from Four Sensors		
	2 axl.	3 axl.	5 axl.	2 axl.	3 axl.	5 axl.	2 axl.	3 axl.	5 axl.	2 axl.	3 axl.	5 axl.
Quartz sensors	6.0	6.0	6.0	3.5	4.0	4.5	6.0	8.0	15.0	6.0	6.0	12.0
Bending plate sensors	8.0	8.0	8.0	3.5	5.0	5.0	10.0	15.0	15.0	6.0	8.0	10.0

## Data Availability

Data are contained within the article.

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
