# Peer review of "Weigh-in-Motion Site for Type Approval of Vehicle Mass Enforcement Systems in Poland"

_sensors, 2023, doi:10.3390/s23229290_

Round 1

Reviewer 1 Report

Comments and Suggestions for Authors

This is an interesting paper about Weigh-in-motion systems and the formulation of requirements for metrological legalisation procedures. Please consider the following comments to improve the quality of the paper:

(1) Line 22: Please highlight a brief conclusion of the research results at the end of the abstract to mention the paper's contribution.

(2) Lines 19 & 20: Grammatical errors should be fixed. Correct "In this paper we describes a unique...".to "In this paper, we describe ...." . Please correct "and a results of" to "and the results of..."

(3) Lines 20 & 379:  There are different duration for the research. Please clarify or correct it: at line 20 "a results of over a year of our research"; at line 379 "assessment of the accuracy of both WIM stations was carried out after a period of almost two years"

(4) Line 465: Considering the experiment of research was in Poland, please add a section about the Limitations of the research. For example, the role of climatic conditions, like temperature, in accuracy of sensors in extreme high/low temperatures, the role vehicle speeds, etc.

Comments on the Quality of English Language

There are some grammatical errors that should be fixed.

Author Response

Our answers to your comments are in the submitted file. Please see to attachment.

Thank you very much for your valuable remarks.

Reviewer 2 Report

Comments and Suggestions for Authors

Dear Authors, I am pleased to have read the results of your research. My assessment is "Accept in present form".

The structure of the article and the form of providing materials meets the highest requirements. The purpose of the study is clearly formulated, the scope and methodology of research are well presented. The negative impact on the measurements of the external environment is fully taken into account. Sufficient quantity and acceptable quality of graphic materials allows you to get a complete picture of the methodology and results of experimental studies. The conclusions (section 6) are formulated clearly and clearly. Table 2 summarizes all the results.

Perhaps I have only two minor wishes for You :

1. Lines 19 and 86 indicate "unique WIM site". Probably, it is still more correct to formulate a "WIM system", but not a site. Please correct the text.

2. In lines 227-230, I ask you to check the dimensions of the Power Spectral Density (PSD) indicator again. It seems to me that there is a dimension error here.

In general, I am impressed with both the article itself and the professional approaches to assessing the weight load on the road surface implemented in Poland. I am sure there is a very serious benefit to the Polish economy from this work.

You are doing a very important thing.

Author Response

(The authors gave the same response as above.)

Reviewer 3 Report

Comments and Suggestions for Authors

1 Literature is poor, the majority of publications are old only a few have been published in the last five years. A lot of reports but not scientific papers are cited. Seven scientific publications cited in the text are written by the first author taking into account that the total number of cited research papers is 10, and self-citation is too high.  

2 In the introduction section authors should focus on the research problem pointing out their contribution.

3 The paper structure is more like a project report, but not a scientific paper. A lot of tasks are presented, but not in the required details to see the novelty and contribution.

4 The last section is the Summary there table with results is presented, but it should be changed to conclusions.

5 My suggestion is to reject the paper with the possibility of resubmitting after rewriting the text according to provided comments.

Comments on the Quality of English Language

English can be improved, there are some strange sentences. For example: 

-In this paper, we present the structure, design and parameters of our research WIM site.

-In Figure 3 a fragment of the actual weighing station equipped with bending plate sensors is visible.

-The installed cameras allow for monitoring of vehicle traffic at the WIM station and recognition of registration numbers of weighed vehicles. 

etc.

Author Response

(The authors gave the same response as above.)

Round 2

Reviewer 3 Report

Comments and Suggestions for Authors

The paper was significantly improved and can be accepted after a minor review. Please improve the introduction from:

The paper [n] presents ...

The paper [n1] presents...

...

Please write it as one paragraph showing the state of the art and pointing out the gap, your paper solves.

Comments on the Quality of English Language

Dear Editor,

Please make a decision if such English is acceptable
